# Learning Interpretable Policies in Hindsight-Observable POMDPs through Partially Supervised Reinforcement Learning

### Abstract

Deep reinforcement learning has demonstrated remarkable achievements across diverse domains such as video games, robotic control, autonomous driving, and drug discovery. Common methodologies in partially observable domains largely lean on end-to-end learning from high-dimensional observations, such as images, without explicitly reasoning about true state. We suggest an alternative direction, introducing the Partially Supervised Reinforcement Learning (PSRL) framework. At the heart of PSRL is the fusion of both supervised and unsupervised learning. The approach leverages a state estimator to distill supervised semantic state information from high-dimensional observations which are often fully observable at training time. This yields more interpretable policies that compose state predictions with control. In parallel, it captures an unsupervised latent representation. These two— the semantic state and the latent state—are then fused and utilized as inputs to a policy network. This juxtaposition offers practitioners a flexible and dynamic spectrum: from emphasizing supervised state information to integrating richer, latent insights. Extensive experimental results indicate that by merging these dual representations, PSRL offers a potent balance, enhancing model interpretability while preserving, and often significantly outperforming, the performance benchmarks set by traditional methods in terms of reward and convergence speed.

## 1 Introduction

The realm of deep reinforcement learning has been transformed through a series of advances that utilize deep neural networks for policy or value function approximations. Dominating the current Reinforcement Learning (RL) landscape are methodologies that view the world as a Markov Decision Process (MDP)— treating sequences of observations as states and often learning policies directly from observable data, such as images translating to controls (Carlucho et al., 2018). Such an end-to-end (E2E) approach has indeed set the benchmark in many domains, like robot navigation (Shi et al., 2019), Atari games (Mnih et al., 2013), human-level driving in Gran Turismo (Wurman et al., 2022), and numerous others.

However, many environments, especially those with high-dimensional sensory inputs, are better modeled as partially observable Markov decision processes (POMDPs). The so-called "true" or "semantic" state in these settings is often a relatively low-dimensional representation which encapsulates semantic information. For example, in autonomous driving, the true state includes critical variables such as the ego-vehicle position, velocity, acceleration, or locations of nearby obstacles. If we are to learn policies that map such semantic information to control, they are naturally more interpretable. However, we do not observe this state directly, but instead have a high-dimensional sensory data stream, such as from camera, LiDAR, and radar. An alternative to typical end-to-end learning that aims to capture some of this intuition has been to learn a low-dimensional latent state representation that is implicit in the raw sensory data (Guo et al., 2023b). However, such representations are typically not interpretable.

Significantly, what all conventional end-to-end approaches, including those which infer latent state, fail to capture is that in many domains true semantic state information is *available during training, even while it is unobservable at execution time*. For example, this is a common setting in robotics and control re-

search (Bansal et al., 2020; Bowman et al., 2017; Dean et al., 2020; Dong et al., 2017; Godbole and Sub-barao, 2019; Kumar et al., 2013; Morton and Kochenderfer, 2017; Rafailov et al., 2021; Tang et al., 2018), and true state information is available by construction in any simulation-based learning environment. Several recent efforts have attempted to exploit this structure to improve the efficiency of reinforcement learning in POMDPs. Lee et al. (2023) consider a closely related model of *hindsight observability* in which information about semantic state is observed at the end of each episode, and develop model-based reinforcement learning approaches that achieve sample complexity that is linear in the number of actions and logarithmic in the size of the transition and observation function spaces. Pinto et al. (2018) and Baisero and Amato (2022) term this problem *asymmetric reinforcement learning* and leverage state information by training critics—but not actors—on states rather than observations. However, while Lee et al. offer a general-purpose algorithm, theirs is a custom model-based approach; further, they assume that once the POMDP has been learned, they can quickly solve it to optimality, making their approach impractical in settings with high-dimensional observations, such as images. On the other hand, while both Pinto et al. and Baisero and Amato offer practical algorithms, these are specific to actor-critic reinforcement learning frameworks. Recently, Baisero et al. (2022) proposed an asymmetric RL algorithm for deep $Q$-learning. However, this approach is specific to DQN-style algorithms (Farebrother et al., 2018; Wang et al., 2016). Thus, asymmetric RL methods are at the moment distinct for $Q$-learning and actor-critic methods, and no practical unified (general-purpose) method exists for making use of known state information in reinforcement learning for POMDPs with high-dimensional observations in an interpretable way. Moreover, existing asymmetric RL approaches ultimately still train policies that are end-to-end in the sense of mapping raw sensory inputs to controls.When these models err it is impossible to know whether the models have not learned useful representations, or they have not learned good policies with respect to those representations. Consequently, concerns about the lack of interpretability of conventional E2E approaches, which yield complex "black-box" policies, apply to these as well.

To address this gap, we introduce a Partially Supervised Reinforcement Learning (PSRL) framework. In PSRL, we assume that training time trajectories include both observation and true state information, whereas only the former is available at execution time. PSRL has two main building blocks: state predictor $g$, which maps (finite) observation history to predicted state, and policy $\pi$, which maps predicted state to action. At training time, both $g$ and $\pi$ are jointly trained using a combination of supervised and reinforcement learning loss. Crucially, we train $\pi$ with *predicted*, rather than actual state as an input, ensuring that the learned policy is robust to prediction errors. These models are interpretable for two reasons. First, it enables designers to develop policies that map actual state to actions independently of raw data, and "plug" these in directly, in addition to the more natural framework of jointly learning predictions and policies as we do in PSRL. This makes the state estimation an "explanation" in the sense defined by Milani et al. (2024). Second, we can now explicitly determine whether the cause of poor behavior due to poor state predictions or poor state-dependent decisions of designed or learned policies. In other words, state prediction provides a kind of counterfactual information, i.e., we can determine what the correct behavior should have been in a given state, so long as it is predicted well. Notably, this notion critically relies on the state definition being semantically relevant. Our framework can't claim these benefits if the state itself is poorly specified. Fortunately, in many practical applications, there is already an extensive literature devoted to specifying meaningful semantics.

At execution time, actions are chosen by composing $\pi$ and $g$, thereby effecting an end-to-end policy that relies only on observations. Nevertheless, this explicit composition preserves a high degree of interpretability, as the policy makes a clear distinction between the prediction of a semantically meaningful state, and actions taken with such predictions as an input. Further, the PSRL framework admits a natural generalization that allows for state to be incompletely observed even at training time, whereby we additionally learn a very low-dimensional representation of the latent part of the state which is concatenated with predicted state as an input to $\pi$. We refer to this as PSRL-$K$, where $K$ is the number of such latent dimensions. This generalization, in turn, induces a space of algorithmic approaches that span from fully end-to-end (ignoring supervised loss in training, with $K$ just the dimension of latent state in end-to-end RL) to PSRL-0, with its added advantage of being interpretable due to the semantic nature of the state predicted by $g$. Thus, varying $K$ induces a tradeoff between interpretability and ability to capture additional information from sensor data that may be relevant for control. In general as $K$ grows the amount of information the policy can use to

make decision increases. This diminishes the counterfactual explabaility efficacy as previously mentioned. Other interpreability techniques such as Selvaraju et al. (2019) would be required to understand how much of the behavior was dependent on the state estimation vs. the latent features.

We investigate the efficacy of PSRL-$K$ for varying $K$ for both deep Q-learning (DQN) and actor-critic (PPO) approaches that instantiate the policy learning architecture and loss, on six domains in OpenAI Gym. Broadly, we find that PSRL-0 is usually, although not always, more sample efficient than state-of-the-art end-to-end counterparts, and the added value of increasing $K$ is typically low. Moreover, our experiments demonstrate that both pre-training the state predictor $g$ or the policy $\pi$—both common ways state information is used in specific applications—tends to perform poorly, in the former case because PSRL makes more efficient use of supervised data, and in the latter case because of fragility of policies learned using true, rather than predicted, state to prediction errors. Finally, our experiments demonstrate interpretability of composite policies learned by PSRL, showing that state prediction errors are typically far lower than unsupervised embeddings learned in conventional end-to-end reinforcement learning paradigms.

**Related Work** Deep reinforcement learning has demonstrated remarkable success in settings in which input information is high-dimensional sensory data, such as images (Arulkumaran et al., 2017; Li, 2017; Mnih et al., 2013; Wang et al., 2018; Wurman et al., 2022). Settings of this kind are naturally modeled as partially-observable Markov decision processes. However, since true state is often observable (albeit, not necessarily obviously) from sensory data, it has proved quite effective to simply treat such high-dimensional input as state, and apply standard deep RL methods to learn end-to-end (that is, raw data to control) policies (Mnih et al., 2013; Wurman et al., 2022). Recently, however, this general tendency has come under scrutiny from two directions. First, in numerous applied settings, such as involving robotics and control applications, it is often natural to decompose end-to-end control into state inference and state-based control (Bansal et al., 2020; Bowman et al., 2017; Dean et al., 2020; Dong et al., 2017; Godbole and Subbarao, 2019; Kumar et al., 2013; Ma et al., 2021; Morton and Kochenderfer, 2017; Rafailov et al., 2021; Tang et al., 2018), although this decomposition is typically ad hoc and domain specific. For example, one may simply learn to predict state from observational data first (i.e., independently of control), and then uses such predictions to synthesize a controller (e.g., using conventional RL). However, such approaches can often fail due to prediction errors (Pinto et al., 2018). The proposed PSRL framework can be viewed as unifying these disparate application-driven approaches, as well as providing a systematic means for studying them. Moreover, the joint supervised and RL training in PSRL overcomes the fragility of many such approaches to prediction errors.

Second, several lines of RL research have observed that one commonly observes true state of a POMDP during training. On the more theoretical side, this has been modeled as *hindside-observability* in POMDPs (HOMDPs), that is, true state can be observed after an action has been taken, or after an episode, during training, inference, or both (Guo et al., 2023a; Lee et al., 2023). This work has demonstrated that observability of state (at least) during training can yield theoretical advantages, but did not yield practical algorithms for high-dimensional settings (e.g., most approaches assume a finite set of states). On the more practical side, several recent approaches proposed *asymmetric RL* (Baisero and Amato, 2022; Baisero et al., 2022; Pinto et al., 2017). The key idea in asymmetric RL is to train a critic that depends directly on state, and an actor that is end-to-end. However, such approaches need to be designed separately for deep $Q$-learning and actor-critic RL frameworks, and still ultimately rely on learning end-to-end actors (for example, in the $Q$-learning context, the goal is to learn an actor which is, approximately, the expectation of the critic $Q$-function). PSRL provides a simple unified framework that is conceptually independent of the particular flavor of model-free RL, making use of supervised information provided by observability of state at training time.

Our work also builds on ideas in current interpretability literature such as Peng et al. (2022) and Luss et al. (2022). These works seek to develop model artifacts that would allow a developer or end user to understand agent actions at each step. Moreover, our work is a practical approach to addressing the anticipatory gap detailed in Finkelstein et al. (2022). The key insight detailed is to formally view interpretability in reinforcement learning as a policy satisfaction problem between the agent and some outside observer (presumed to be a human who needs the explanation). We address this by mapping observations directly to a semantic state.

Our work also connects to the representation learning literature (Bengio et al., 2013). These works seek to understand the theoretical and practical impacts of the way networks use underlying representations of their input data, and the way these representations are leveraged by downstream tasks. He et al. (2023) demonstrated that for optimal policies learned through DQN, the primary Q network needs to learn distinguishable representation of the underlying input data. Le Lan et al. (2023) looked at how the latent representations change when auxiliary objectives are incorporated into the training process of temporal difference learning. Their work hypothesises the potential case where the representation is parameterized by a neural network, as we have done here. Of particular relevance is Wang et al. (2023), which attempts to solve the POMDP problem using VAE constructed representations over observations. While this method can be useful for comparing underlying abstract belief states from actions, it does not capture the intreptability notions detailed here.

Finally, our work is loosely related to model-based reinforcement learning, such as DREAMER, which also uses supervised techniques (Jin et al., 2019; 2021; Okada and Taniguchi, 2021; Wu et al., 2023; Yang and Wang, 2019). However, in model-based RL, supervised techniques focus primarily on learning dynamics and rewards, whereas the propose PSRL framework uses supervision as a means to take advantage of observed state at training time in learning how to act in POMDPs.

## 2 Preliminaries

**Partially Observable MDPs** A partially-observable Markov decision process (POMDP) is characterized by a tuple: $\{S, A, T, O, \Omega, r, \gamma, \rho\}$, where $S$ is the state space, $A$ the action space, $T$ the transition function where $T(s'|s, a)$ is the probability distribution over $s' \in S$ given true state $s$ and action $a$, $\Omega$ the set of observations, $O$ the observation function with $O(o|s)$ the distribution of observations $o \in \Omega$ given true state $s \in S$, $r(s, a)$ the reward function, $\gamma \in [0, 1)$ the temporal discount factor, and $\rho$ the distribution over initial state $s_0$. Throughout, we assume that $S \subseteq \mathbb{R}^n$ and $\Omega \subseteq \mathbb{R}^m$, and that typically $m \gg n$, that is, the dimension of the raw sensory observation space is far greater than the dimension of the true state $s$, which furthermore is semantically meaningful (for example, position, velocity, and acceleration of a car). We allow $A$ to be either finite or infinite and multi-dimensional.

In general, a policy in a POMDP can be a function of an arbitrary observation history. However, in practice it is typical to keep track of only a finite sequence of observations (say, the last $L$). For simplicity, and since an observation $o_t$ can embed such a finite sequence, we aim to learn policies $\pi(o_{t-i}, ..., o_t)$ that map an observation sequence $o_{t-i}, ..., o_t \in \Omega$ to an action $a \in A$. Formally, an optimal policy of this form solves the following problem:

$$\max_\pi \mathbb{E}\left[\sum_{t=0}^\infty \gamma^t r(s_t, a_t)|a_t = \pi(\cdot)\right],$$

where the expectation is with respect to $\rho$ as well as the distribution of trajectories induced by a policy $\pi$ and transition function $T$.

**Deep Reinforcement Learning** When the POMDP model is given and the state, observation, and action sets are finite and relatively small, it is possible to compute an optimal policy relatively effectively Shani et al. (2013). However, in many problem domains we are not given the POMDP model, but instead have an environment—commonly, a simulator—that we can experiment with by taking actions $a$ and observing rewards $r(s, a)$. In such cases, reinforcement learning (RL) provides a general framework for *learning* how to act, that is, for learning a policy $\pi$ from experience. Modern approaches for reinforcement learning with states represented by real vectors rely on function approximation in which the value function, policy, or both are represented using deep neural networks. While most such approaches have been developed for MDPs (i.e., where state $s$ is fully observable at both training and execution time), a common adaptation to POMDPs is to replace states $s$ with a sequence of observations $o_{t-1}, ..., o_t$, effectively treating this sequence as state, and apply standard RL methods as if it where an MDP; prototypical examples of this are Mnih et al. (2013) and Wurman et al. (2022).

We consider two classes of reinforcement learning algorithms. The first involves variants of $Q$ learning that are most common when the set of actions $A$ is relatively small. The second class are policy gradient methods, including actor-critic approaches, most common when the set of actions is large or infinite.

In relatively general terms, deep reinforcement learning algorithms can be viewed as minimizing a composite loss function comprised of up to three elements: 1) critic loss $\mathcal{L}_c(\theta_c)$, 2) actor loss $\mathcal{L}_a(\theta_a)$, and 3) entropy $\mathcal{H}_e$:

$$\mathcal{L}_{RL}(\theta_c, \theta_a) = \mathbb{E}_{r,s,a,s',a'}[\mathcal{L}_c(\theta_c) + \alpha_1 \mathcal{L}_a(\theta_a) + \alpha_2 \mathcal{H}]. \tag{1}$$

In the case of deep Q-learning, $\alpha_1 = \alpha_2 = 0$, and in the most basic variant of deep Q network (DQN) Mnih et al. (2015), we learn the parametric action-value function $Q_{\theta_c}(s, a)$ using loss $\mathcal{L}_c(\theta_c) = (\bar{Q} - Q_{\theta_c}(s, a))^2$, where $\bar{Q} = r + \max_{a'} Q_\theta(s, a')$. Variations, such as double DQN (DDQN), involve a distinct Q network as a target in $\bar{Q}$ van Hasselt et al. (2015). In the case of actor-critic methods, one commonly learns a value function $V_{\theta_c}(s)$ as the critic. In particular, in this case it is common to define the advantage function $A(s, a) = (\bar{V} - V_{\theta_c})$, where $\bar{V} = r + V_{\theta_c}(s')$, and the critic loss is just $\mathcal{L}_c(\theta_c) = A(s, a)^2$, while the actor loss is $\mathcal{L}_a(\theta_a) = A(s, a) \log \pi_{\theta_a}$, where $\pi_{\theta_a}$ is the policy (often represented as a neural network), as in the case of approaches as such advantage actor critic (A2C) Mnih et al. (2016), or

$$\mathcal{L}_a(\theta_a) = -A(s, a) \min \left\{ \frac{\log \pi_{\theta_a}}{\log \pi_{\theta_{old}}}, \right.$$
$$\left. \text{clip}(\frac{\log \pi_{\theta_a}}{\log \pi_{\theta_{old}}}, 1 - \eta, 1 + \eta) \right\},$$

as in PPO Schulman et al. (2017). Finally, the entropy term $\mathcal{H}$ aims to ensure that the actor $\pi$ remains stochastic during training to ensure exploration.

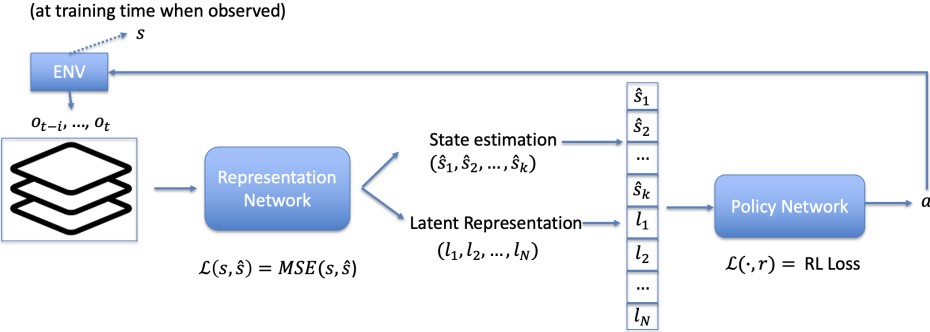

Figure 1: The PSRL-$K$ framework.

## 3  Approach

In a broad array of domains, it is reasonable to assume that even though the domain is partially observable, true state is observed *during training*, though not at execution time. For example, any simulation, by construction, must generate both true state and observations, while only emitting observations to mimic whatever domain is being simulated. Moreover, common robotic settings involve highly instrumented training runs (besides training using simulations) in which one can indeed closely approximate whatever actual state information is important. Formally, we assume that training trajectories $\tau = (s_0, o_0, a_0, s_1, o_1, a_1, \ldots, s_t, o_t, a_t, \ldots)$ include *both* observations $o_t$ and states $s_t$ at each time step $t$.

We propose to leverage knowledge of true state at training time by combining reinforcement learning, which learns how to act, with supervised learning, which learns to predict state $s$ from observations $o$. Let $g_\phi(o_{t-i}, \ldots o_t)$ denote a parametric model that predicts state given an observation sequence of the last

$i+1$ observations. In general, $g_\phi$ can be either a point prediction or a distribution. Here, we assume it makes a point prediction of a state $s$ given an observation sequence to predict $s$ at time $t$ ie the last observation in the sequence corresponds to the predicted state. Recall that policy $\pi_{\theta_a}(\cdot)$ takes an observation sequence sequence (potentially a single observation) as an input. We refer to this as an *end-to-end* policy. The key idea, which is either implicitly or explicitly common in numerous particular cases , but has not previously been systematically investigated on its own, is to have a policy with an architecture that explicitly composes state prediction with decision. Formally, let $\tilde{\pi}_{\theta_a}(s)$ be a policy that maps the *true* state $s$ (albeit, unobserved at decision time) to actions. We can then define $\pi(o_{t-i}, ..., o_t) = \tilde{\pi}_{\theta_a}(g_\phi(o_{t-i}, ..., o_t))$.

If both $\tilde{\pi}$ and $g$ are differentiable, we can train the resulting policy end-to-end using the conventional RL loss $\mathcal{L}_{RL}(\theta_c, \theta_a)$ in Equation (1). And, in conventional RL settings, there is little else to do. However, when we observe state $s$ during training, we can do more. Recent asymmetric RL approaches take advantage of such information by learning a critic value or action-value ($Q$) function as a function of true state, rather than observation, with the actor (policy) Baisero and Amato (2022); Baisero et al. (2022); Pinto et al. (2018). However, actor policies in these methods are still learned end-to-end. We suggest that this is a missed opportunity for two reasons: first, because it does not take full advantage of the knowledge of true state at training time, and second, because distinct approaches are needed for actor-critic and DQN settings. We propose *partially-observable reinforcement learning (PSRL)* as a general framework for RL that leverages knowledge of true state of a POMDP during training. In its most basic variant, we jointly train $\tilde{\pi}$ and $g$ using a combination of supervised loss (for $g$) and RL loss (for both):

$$\mathcal{L}_{PSRL}(\theta_c, \theta_a, \phi) = \mathcal{L}_{RL}(\theta_c, \theta_a) + \beta \mathcal{L}_S(\phi), \qquad (2)$$

where $\mathcal{L}_S(\phi)$ is the supervised (e.g., cross-entropy) loss and $\beta$ a parameter that determines the relative weight of the supervised loss to RL loss. Note that since the policy architecture composes the state-level policy $\tilde{\pi}$ and predictions $g$, effectively both the actor and the critic in actor-critic method is still trained with respect to observation input $o$, unlike asymmetric actor-critic, in which the critic takes true state as an input. A corresponding asymmetric variation of PSRL is straightforward. A key advantage of PSRL over asymmetric actor-critic, however, is its simplicity and generality; in particular, it applies equally directly to both $Q$ learning and policy gradient methods.

An important limitation of the PSRL approach is that it assumes decodability of the true state from the observation (Mhammedi et al., 2020), that is, that each observation is associated with a unique state.

To address this, we propose a generalization of PSRL, PSRL-$K$, in which the policy $\tilde{\pi}$ takes as input (predicted) state $s$ along with $K$ latent variables trained solely using the RL loss. Formally, let $\tilde{\pi}(s, z)$, where $z$ represents a latent low-dimensional observation embedding, and let $z = h_\psi(o)$ be a neural network architecture that captures this embedding. The full policy architecture is then $\pi(\cdot) = \tilde{\pi}_{\theta_a}(g_\phi(o), h_\psi(o))$, still trained using the PSRL loss in Equation (2). We provide a schematic illustration of PSRL-$K$ in Figure 1.

Note that PSRL-$K$ generalizes both PSRL and end-to-end RL. In the former case, we omit the dependence on $h_\psi$ (equivalently, set $K = 0$), while in the latter case, we omit the dependence on $g_\phi$ (and, therefore, the supervised part of PSRL loss). Consequently, it enables us to modulate between the two extremes, which our experiments show can be beneficial.

Finally, the proposed joint supervised and RL training in PSRL and its generalization is unlike a rather natural idea common in applied settings, where we train a state predictor $g$ and independently a state-conditional policy $\tilde{\pi}$, and compose these at decision time. However, this approach is sensitive to state prediction errors. An improvement is to first pre-train $g$, and then train $\tilde{\pi} \circ g$ using either RL or PSRL loss. We explore these alternatives in the experiments below.

## 4 Experiments

We evaluate the performance of PSRL-$K$ on several common benchmark environments from OpenAI Gym, comparing to state of the art end-to-end and asymmetric actor critic approaches, as detailed below. In three environments featuring small action sets, we evaluate the implementation of PSRL-$K$ in deep $Q$-learning-

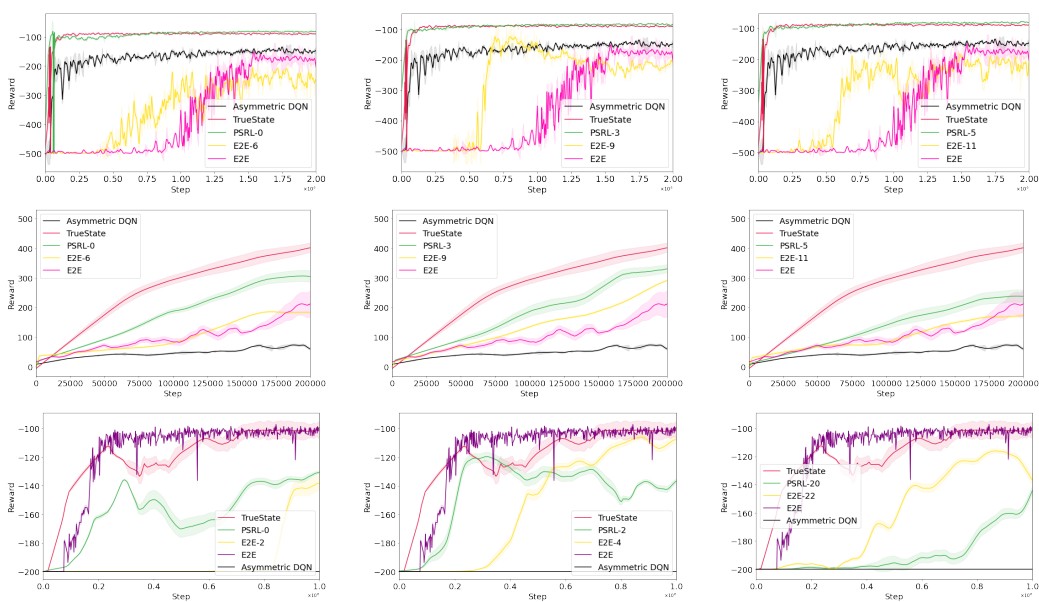

Figure 2: Experiments in Acrobot (top row), Cart Pole (with finite action sets; middle row), and Mountain Car (bottom row), in finite-action environments using DDQN approaches.

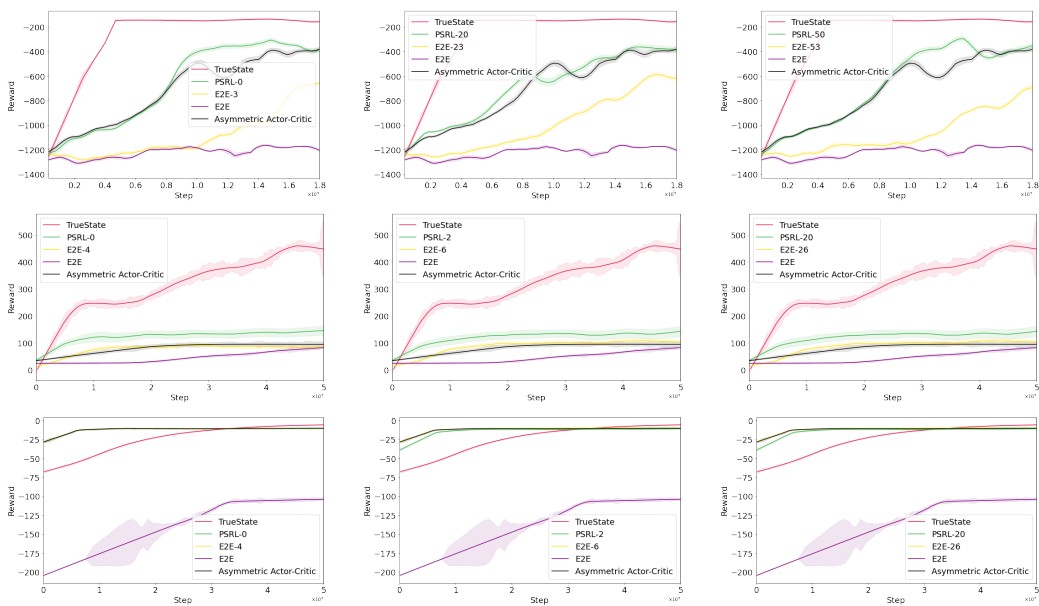

Figure 3: Experiments in Pendulum (top row), Cart Pole (with continuous action sets; middle row), and Reacher (bottom row), in continuous-action environments using PPO approaches.

based approaches (specifically, double deep $Q$-network (DDQN) (van Hasselt et al., 2015)). In three others that feature continuous actions, we combine PSRL-$K$ with PPO.

## 4.1 Experiment Setup

**Environments**  We consider five OpenAI Gym environments: Acrobot, Cart Pole, Mountain Car, Reacher, and Pendulum (OpenAI, 2023a;b;c;e;d). Of these, Acrobot and Mountain Car have finite action sets, Reacher and Pendulum involve continuous actions, and we used two versions of Cart Pole, one with finite and another

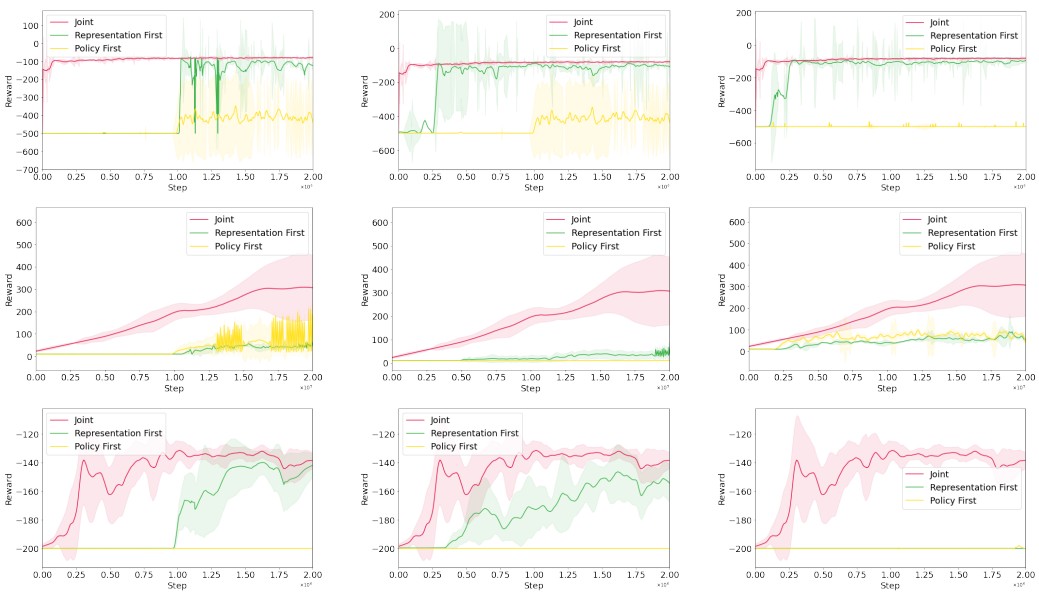

Figure 4: Experiments comparing PSRL-0 (joint RL and supervised) learning with either representation first or policy first approaches. Top row, left-to-right: Acrobot 50% pretrained, Acrobot 25% pretrained, Acrobot 10% pretrained. Middle row: Cartpole 50% pretrained, Cartpole 25% pretrained, Cartpole 10% pretrained. Bottom row: Mountain Car 50% pretrained, Mountain Car 25% pretrained, Mountain Car 10% pretrained.

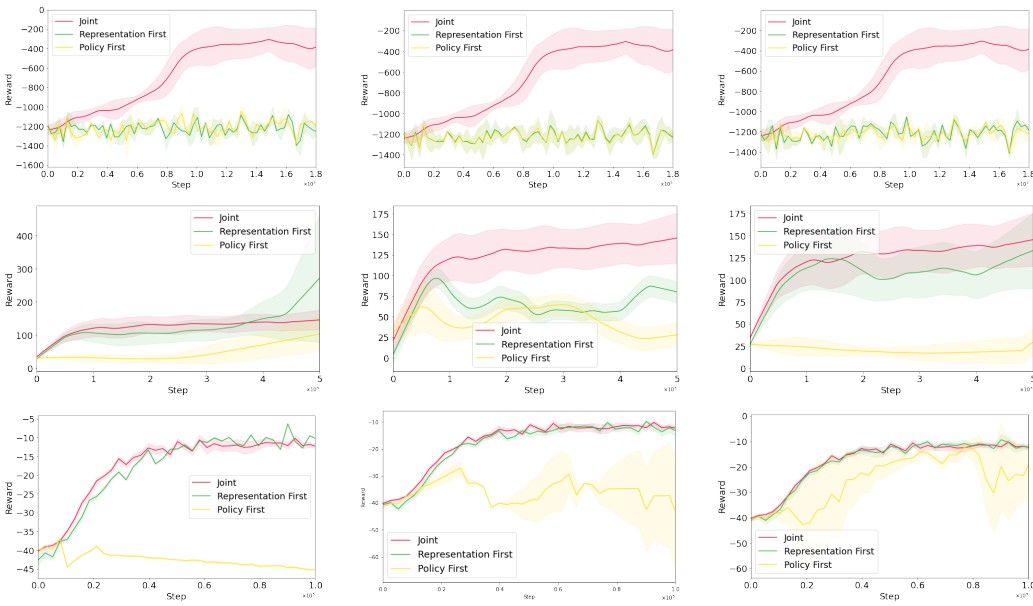

Figure 5: Experiments comparing PSRL-0 (joint RL and supervised) learning with either representation first or policy first approaches. Top row, left-to-right: Pendulum 50% pretrained, Pendulum 25% pretrained, Pendulum 10% pretrained. Middle row: Continuous Cartpole 50% pretrained, Continuous Cartpole 25% pretrained, Continuous Cartpole 10% pretrained. Bottom row: Reacher 50% pretrained, Reacher 25% pretrained, Reacher 10% pretrained.

with continuous action sets. In Acrobot, (true) state is a 6-dimensional vector providing information about the two rotational joints and two links. In Mountain Car, state is represented by a 4-dimensional vector, comprising the cart position, cart velocity, pole angle, and pole angular velocity. In Pendulum state is a

3-dimensional vector, representing the cosine and sine of the pendulum's angle and its angular velocity. In Cart Pole is a 4-dimensional vector, comprising the cart position, cart velocity, pole angle, and pole angular velocity. Finally, in Reacher the state is given by an 11-dimensional vector comprising the cosine and sine values for the angles of both arms, the coordinates of the target, the angular velocities of the arms, and the vector between the target and the reacher's fingertip. In all cases, observations $o$ are short sequences of 2D images simulated by OpenAI Gym.

**Baselines** Our first baseline is largely for calibration, and assumes that the environment is fully observable. This *true state* baseline simply learns the policy $\pi(s)$ using either DDQN (in finite-action settings) or PPO (in continuous-action settings). The second baseline is end-to-end (*E2E*), for which we learn policies $\pi(o)$ directly mapping image inputs to control using either DDQN or PPO as conventionally done. For each environment, we use the best performing E2E baseline available. To ensure a fair comparison with PSRL-$K$, we additionally provide results for E2E-$K + n$, where the architecture of $\pi(o)$ mirrors that of PSRL-$K$, except the policy is learned using solely the RL loss. Finally, we compare with a state-of-the-art asymmetric DQN (*ADQN*) (Baisero et al., 2022) in the finite-action settings and asymmetric PPO (*APPO*) (Baisero and Amato, 2022) in the continuous-action settings.

## 4.2 Results

**Q-Learning Settings** Our first set of experiments considers the DDQN framework, comparing PSRLwith the baselines. The results are presented in Figure 2. Consider first the top row, corresponding to Acrobot. In this setting, PSRLis essentially as effective as learning in its fully observable counterpart. In contrast, both variants of end-to-end learning considerably slower, even as they ultimately approach a near-optimal policy. Finally, we observe that ADQN significantly outperforms E2E approaches, but remains clearly below PSRLin terms of performance.

The second row of Figure 2 presents DDQN results for the finite-action Cart Pole environment. Here, PSRLagain exhibits a clear advantage over end-to-end approaches as well as asymmetric DQN, although in this domain there is a clear advantage in the knowledge of true state. Notably, in both the Acrobot and Cart Pole domain, there appears to be little advantage to PSRL-$K$ for $K > 0$, with PSRL-0 already exhibiting strong performance.

Finally, the last row of Figure 2 presents the results for Mountain Car. In this domain, we find that end-to-end performs extremely well, essentially no different from using the true state. PSRL-0 is tangibly worse, but in this case PSRL-2 performs better.

**Actor-Critic Settings** Next, we consider settings with continuous actions, combining PSRLand baselines with the PPO actor-critic approach. The results are provided in Figure 3. In the Pendulum environment (top row), both asymmetric actor-critic and PSRLoutperform end-to-end approaches by a large margin, although learn tangibly slower than true-state PPO. Moreover, PSRL-0 learns somewhat faster than asymmetric actor-critic. In the Cart Pole environment (bottom row), the advantage of PSRL and asymmetric methods over end-to-end is smaller. Indeed, in this case, asymmetric actor critic no longer outperforms one of the end-to-end baselines. PSRL, however, maintains an advantage over both. In both of these domains, PSRL-0 performance is comparable to PSRL-$K$ for several values of $K$; that is, no added value is provided by including latent features. Finally, in the Reacher environment, PSRLand APPO are both comparable and actually learn somewhat faster than the approach which knows the true state (perhaps because of the implicit increase in exploration that results from imperfect state predictions), whereas end-to-end approaches are considerably worse.

**Ablation Experiments** Now, we consider two natural ablations of PSRL. The first ablation involves pretraining state prediction (representation network) $g_\phi$ using only supervised loss for the first $T$ iterations. Thereafter, we freeze the representation network $g_\phi$, and train only the policy network $\tilde{\pi}(g_\phi(o))$ using conventional RL loss. This roughly mirrors the typical way that state information in reinforcement learning is used in applied settings, such as robotic control (Bansal et al., 2020; Godbole and Subbarao, 2019; Morton and Kochenderfer, 2017; Tang et al., 2018). We refer to this as *representation-first PSRL*. The second ablation first pretrains a policy mapping true state to actions for the first $T$ iterations using RL loss. Thereafter, we freeze the policy network $\tilde{\pi}$, and train the state prediction $g_\phi$ using supervised loss. We refer to this

as *policy-first PSRL*. This corresponds to an effective decomposition of prediction and policy training, also common in applied settings.

The results are provided in Figure 4 for discrete action environments and Figure 5 for continuous action environments. We can observe that neither the representation first nor the policy first baselines are as effective as PSRLin the majority of cases. However, there are a few interesting exceptions. While the policy first baseline is nearly always poor, representation first is effective in Continuous Cart Pole and Reacher, indistinguishable from PSRLin the latter, and actually outperforming it under one configuration in the former. Interestingly, these exceptions are only in the PPO setting, whereas the joint representation and policy training in PSRLappears uniformly more advantageous than these variations in the DQN setting.

**Wall Time Experiments** We find that our method is comparable or considerably faster than E2E baselines. Below are training times for continuous domains are over 10000 time steps and discrete action over 100000 time steps.

Table 1: Comparison of Truestate, E2E, and PSRL timings for different environments.

| Truestate (s) | E2E (s) | PSRL (s) | Env |
|---|---|---|---|
| $5491 \pm 191$ | $85343 \pm 851$ | $44023 \pm 6069$ | Cartpole |
| $3332 \pm 69$ | $70643 \pm 8721$ | $55721 \pm 1153$ | Acrobot |
| $8412 \pm 19$ | $123892 \pm 36304$ | $45008 \pm 1997$ | Mountain Car |
| $67 \pm 2$ | $1052 \pm 43$ | $2930 \pm 54$ | Continuous Cartpole |
| $69 \pm 0.1$ | $1292 \pm 3$ | $1292 \pm 3$ | Pendulum |
| $66 \pm 0.1$ | $12985 \pm 179$ | $13528 \pm 87$ | Reacher |

Table 2: Comparison of PSRL-0 and E2E test mean squared error for different environments.

| Environment | PSRL-0 | E2E |
|---|---|---|
| Acrobot | $0.78 \pm 0.27$ | $16260 \pm 7435$ |
| Mountain Car | $0.0003 \pm 0.0001$ | $337 \pm 37$ |
| Cartpole | $0.20 \pm 0.005$ | $56 \pm 5.6$ |
| Reacher | $0.30 \pm 0.13$ | $0.88 \pm 0.13$ |
| Cont Cartpole | $0.05 \pm 0.001$ | $805 \pm 146$ |
| Pendulum | $1.04 \pm 0.18$ | $134 \pm 21$ |

**Interpretability** Finally, we consider the issue of interpretability, which we quantify simply as the quality of true (semantic) state estimation, comparing PSRL (that is, the predictions by $g$) with E2E with respect to the embedding of the identical dimension. One may expect that since E2E approaches typically learn to be quite effective, they do so by learning a good estimate of true state despite not having any explicit supervision. Our results in Table 2 dispel this notion, showing that while PSRL yields highly accurate state predictions, latent embedding in E2E is entirely uninterpretable, with distance to true state often many orders of magnitude higher than that for PSRL.

## 5   Conclusion

The PSRL framework reveals its effectiveness as a compelling alternative for addressing the challenges posed by important classes of partially observable domains, striking a balance between performance and interpretability. Through the fusion of supervised and unsupervised learning components, PSRL leverages a state estimator to distill semantic information from high-dimensional observations, providing interpretability in the inferred state. The dynamic spectrum created by the juxtaposition of the semantic and latent states offers practitioners a flexible approach, allowing them to navigate between emphasizing supervised state information and integrating richer latent insights. Our experimental results underscore the versatility and efficacy of PSRL across diverse domains, commonly outperforming traditional methods in terms of reward and convergence speed.

In the broader context of AI, where interpretability and performance often present conflicting objectives, PSRL emerges as a promising methodology. As we continue to advance our understanding and capabilities in complex, partially observable environments, further research and refinement of PSRL could contribute significantly to the evolution of deep reinforcement learning paradigms in practical applications. Our findings suggest that PSRL represents an important step towards achieving a more interpretable and effective approach to deep reinforcement learning.

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
