# OpenReview forum: "Learning Interpretable Policies in Hindsight-Observable POMDPs through Partially Supervised Reinforcement Learning"
_TMLR — Withdrawn by Authors_

### Review · Reviewer_aM26 · 2024-07-20

**Summary Of Contributions:**

The authors introduce the partially supervised reinforcement learning (PSRL) framework that utilizes supervised learning to estimate the state information, and in addition, it also leverages unsupervised learning to learn latent representations of the state -- both the state estimate and the latent representation together feed into the policy network that is trained to predict optimal actions.

The proposed general technique is applied to various RL algorithms and shown to significantly out-perform baselines.

**Audience:**

Yes

**Broader Impact Concerns:**

I don't see any broader impact concerns.

**Claims And Evidence:**

Yes

**Requested Changes:**

Rather questions:

1. Lot of the discussion is around the state estimate and the latent variables feeding into the policy network -- what is the policy network for Q-learning? Are you suggesting to use this as input to the Q-network?

2. I do not completely understand how the true state is observable during "training" but not during "inference"? In RL, what ever you observe during training should be the same as in inference, from my understanding?

3. What is the loss function for the latent variables? It is not clear what those variables are and how they're learnt.

**Strengths And Weaknesses:**

Strengths:

It is great to see a seemingly simple idea have such a nice impact on performance. I really like the somewhat (but not too much) obvious / intuitive idea, and see it perform so well.

In addition, the experiments seem thorough.

Weaknesses:

The biggest weakness in my opinion is clarity -- it may be my fault that I do not understand the details 100%, but I also think more effort needs to be put in to make things more clear. See questions that I have asked below.

Second, though this is a really nice idea, I am not sure if it is in scope for TMLR journal -- seems to be a very good paper that contains detailed experiments, but there's no theory or super novel about this work.

---

> ### Author Response · Authors · 2024-07-23
> **Question about process**
>
> Apologies for a silly question but it is our first time submitting here. Should we be waiting for the other reviewers review before responding? Or are we ok to respond? Additionally, when should we be making changed to the text? After all reviews or during?

---

> > ### Comment · Action_Editor_hJiS · 2024-07-24
> >
> > I would recommend waiting until all reviews are submitted; your responses and edits will probably want to account for the other reviewers' comments.

---

### Review · Reviewer_NnrN · 2024-07-27

**Summary Of Contributions:**

The submitted manuscript described a method to learn policies for partial observable environments with the assumption that true state can be observed during training. The authors further argue that the learned policy can have certain interpretability.

**Audience:**

No

**Broader Impact Concerns:**

N/A.

**Claims And Evidence:**

No

**Requested Changes:**

Please follow the weaknesses part.

**Strengths And Weaknesses:**

## Weaknesses:
* The novelty is limited. The authors just proposed to add an auxiliary loss for the standard end-to-end training when the hidden state is observable during training.
* The formulation is rather confusing. I would claim that for most of the scenarios with partial observability, observation sequence itself will not be sufficient for identifying the states. Action sequence is also needed.
* Lack of comparison: I believe dreamer-style algorithm can do even better on the tested setting even without observing the hidden state during training, but the authors didn't include them. The authors only include several old baselines like DDQN and PPO.
* Limit demonstration: OpenAI Gym environments is pretty simple and I expect there are more complicated tasks including vision inputs or more realistic robot learning settings like Meta-World.
* Vague claim: The interpretability only follows from a MSE on the latent state, which is definitely misleading. How can we know the end-to-end methods not learn a rotated representation of the ground truth state?

---

### Review · Reviewer_rphj · 2024-07-28

**Summary Of Contributions:**

The paper proposes PSRL, a combination of a supervised learning loss which leverages ground truth state for POMDP with RL losses. The paper naturally proposes PSRL-K with parameter K that modulates between two extremes. The paper empirically compares PSRL-K with other baseline methods and shows improvements in certain settings.

**Audience:**

Yes

**Claims And Evidence:**

No

**Requested Changes:**

=== **Presentation** ===

The paper can benefit significantly from a better clarity of presentation. The paper uses too many words to describe the algorithmic procedure, while better notations, algorithm box and more clear flow diagram would help. The paper lacks in clarity in presenting the most important ideas in the paper, for example the notation $\mathcal{L}_s$ which denotes the state dependent supervised learning loss that makes up the main contribution of the paper, is never properly introduced. Indeed, the notation $\mathcal{L}_s$ is never defined in the manuscript, except for a short word description below Eqn 2.

The paper can also significantly cut down the background material and align equations better such as the PPO loss.

The diagram Fig 1 never clearly explains what is $\hat{s}$ and $l$, and there should be a better caption.

=== **Ground truth state** ===

It is rarely the case that one has access to ground truth state in POMDP, hence making the experimental setup a bit questionable and lacking motivation. An interesting case would be if one can construct an estimate of the ground truth state in the hindsight, such as using the entire trajectory's information or the entire dataset, to construct a ground truth estimate, and see if the proposed method would help. In this case, I would guess asymptotically it would become helpful as more data is collected, but I am less sure if during the initial stage when the state is poorly estimated, whether the proposed approach would speed up learning.

=== **Empirical result** ===

It is underwhelming that ultimately the idea is only tested on a few basic control tasks which are converted to POMDP in an artificial way. Note that these tasks allow for accessing ground truth state but this is not the case in general, for more challenging control tasks in deep RL. The experimental test suite is quite weak for a deep RL paper.

In Figure 2-3, rows are different envs, how about columns? This should be clear from the caption.

=== **Interpretability** ===

The paper has little paragraphs on interpretability. Since the proposed algorithm assumes access to ground truth state during training and makes use of such information, it is no surprise that the resulting representation can be more "interpretable" in that they stay closer to the ground truth, which is inherently more interpretable. The gain in interpretable representation stems from access to privileged information, not a more technically novel algorithm. Gaining access to the ground truth state is the most challenging bit of the pipeline, which current paper does not address.

**Strengths And Weaknesses:**

The paper's strength lies in that it studies the effect of having access to ground truth state for POMDP at training time, what algorithmic techniques might help leverage such privileged information. This might be somehow of interest to POMDP community and practitioners who might figure out how to access ground truth state in hindsight.

The paper's main weakness is that it is not clear if the technical contribution is solid enough in this work. Having access to ground truth information at training would naturally improve model performance, the paper proposes one way to leverage such information but other alternatives are not ablated carefully. It is also not clear if such a setting is interesting in the first place, rendering a lack of motivation for this work. The technical section also does not provide strong enough evidence in challenging enough domains, and constraining mostly to low-dimensional control tasks which are made artificially partially observable.

---

### Note · Authors · 2024-07-29

**Comment:**

We thank the reviewers for their time. We are going to withdraw this paper.

**Withdrawal Confirmation:**

I have read and agree with the venue's withdrawal policy on behalf of myself and my co-authors.